# Role of High Voltage-Gated Ca^2+^ Channel Subunits in Pancreatic β-Cell Insulin Release. From Structure to Function

**DOI:** 10.3390/cells10082004

**Published:** 2021-08-06

**Authors:** Petronel Tuluc, Tamara Theiner, Noelia Jacobo-Piqueras, Stefanie M. Geisler

**Affiliations:** Centre for Molecular Biosciences, Department of Pharmacology and Toxicology, University of Innsbruck, Innrain 80/82, 6020 Innsbruck, Austria; tamara.theiner@uibk.ac.at (T.T.); noelia.jacobo-piqueras@uibk.ac.at (N.J.-P.); stefanie.geisler@uibk.ac.at (S.M.G.)

**Keywords:** voltage gated calcium channel, beta cell, insulin, diabetes

## Abstract

The pancreatic islets of Langerhans secrete several hormones critical for glucose homeostasis. The β-cells, the major cellular component of the pancreatic islets, secrete insulin, the only hormone capable of lowering the plasma glucose concentration. The counter-regulatory hormone glucagon is secreted by the α-cells while δ-cells secrete somatostatin that via paracrine mechanisms regulates the α- and β-cell activity. These three peptide hormones are packed into secretory granules that are released through exocytosis following a local increase in intracellular Ca^2+^ concentration. The high voltage-gated Ca^2+^ channels (HVCCs) occupy a central role in pancreatic hormone release both as a source of Ca^2+^ required for excitation-secretion coupling as well as a scaffold for the release machinery. HVCCs are multi-protein complexes composed of the main pore-forming transmembrane α_1_ and the auxiliary intracellular β, extracellular α_2_δ, and transmembrane γ subunits. Here, we review the current understanding regarding the role of all HVCC subunits expressed in pancreatic β-cell on electrical activity, excitation-secretion coupling, and β-cell mass. The evidence we review was obtained from many seminal studies employing pharmacological approaches as well as genetically modified mouse models. The significance for diabetes in humans is discussed in the context of genetic variations in the genes encoding for the HVCC subunits.

## 1. Introduction

Plasma glucose concentration is controlled by several hormones produced and secreted by the cells of the pancreatic islets of Langerhans. The β-cells are the major cellular component (~70%) of the pancreatic islet and secrete insulin, the only hormone capable of lowering plasma glucose concentration [1,2]. The insulin counter-regulatory hormone, glucagon, is secreted by the pancreatic α-cells, which account for ~20% of the total islet cells. Intercalated between α- and β-cells and in close functional connectivity with the β-cells, the pancreatic δ-cells (5%) secrete somatostatin, a hormone critical for islet paracrine modulation [3,4]. While human islets display a mosaic pattern of α-, β-, and δ-cells, mouse islets contain a core of β-cells surrounded by an α-cell mantle [5]. Despite these anatomical differences, which have been discussed in detail elsewhere [6,7,8], numerous studies have shown that the molecular mechanisms of glucose-induced insulin secretion (GIIS) are fairly well conserved between mouse and men. In low plasma glucose concentrations, the K^+^ equilibrium potential sets the mouse β-cell resting membrane potential between −80 mV and −70 mV [6]. Following the rise in plasma glucose concentration, glucose uptake in the pancreatic β-cells and subsequent mitochondrial metabolism result in increased ATP/ADP ratio. This inhibits the activity of the ATP-sensitive K^+^ channels (K_ATP_) [9,10,11,12,13]. It has been postulated that β-cells must be equipped with a “background inward current”, which depolarizes the membrane when K_ATP_ channel activity is low. Cl^−^-permeable leucine-rich repeat containing protein Swell1 [14] and various members of the nonselective cation channels transient receptor potential (TRP) [6] are in this regard the most promising candidates. As soon as the membrane potential reaches the depolarization threshold of the voltage-gated Na^+^ and Ca^2+^ channels (between −60 mV and −50 mV), their rapid activation leads to an increase in the electrical activity in a glucose-dependent manner [6]. In glucose concentrations >6 mM the β-cell electrical activity is characterized by trains of action potentials (AP) on top of a depolarizing plateau alternating with electrically quiet and hyperpolarized intervals [6,15,16,17]. Raising the glucose concentration to ~10 mM increases the frequency and duration of the AP-trains as well as the AP frequency during a train. Above 15 mM glucose, the β-cells display an almost continuous electrical activity. Most excitable cells are capable of spontaneous or even induced electrical activity over a very narrow interval of the resting membrane potential. In contrast, β-cells express in the membrane a whole repertoire of voltage-gated ion channels perfectly suited for the generation and maintenance of spontaneous activity over a very wide range of membrane potentials. This is because the ion channel complement contributing to β-cell activity changes rapidly with the change in plasma glucose concentration. The voltage-gated ion channels reside in three main conformational states dependent on the resting membrane potential, e.g., resting, activated, and inactivated state [18,19,20,21,22,23,24]. The transition between the three states as well as the state in which each channel finds itself at any given time is voltage- and time-dependent. Normally, when the resting membrane potential sits at -80mV, most ion channels are in the resting, ready-to-open state. The increasing membrane potential induces conformational changes that lead to channel activation. During the membrane depolarization, most ion channels undergo intrinsic time-dependent conformational rearrangements that drive the channel into inactivation. The transition from inactivated to resting state (closed but ready to open again) occurs at resting membrane potentials and is time-dependent. These three important ion channel conformations and activity states dictate at any given time and in a membrane potential-dependent manner the percentage of ion channels available for the generation and maintenance of the electrical activity in any excitable cell. As mentioned above, in pancreatic β-cells, the resting membrane potential is glucose-dependent and therefore also the ion channel availability. Most mouse β-cells express voltage-gated Na^+^ channels (Na_V_) that inactivate at very hyperpolarized potentials [15,16]. In low glucose concentrations, the voltage-activated Na^+^ currents contribute to AP generation, a notion substantiated by the fact that tetrodotoxin (TTX) application reduces the AP frequency and amplitude as well as insulin release [6]. Under high glucose concentrations, the increasing ATP/ADP ratio leads to an almost complete K_ATP_ channel block increasing the membrane potential during an AP train above the steady state inactivation voltage of the Na_V_ channels. In consequence, TTX application had minimal effects on AP properties. However, the application of L-type Ca^2+^ channel blockers completely inhibited β-cell electrical activity, demonstrating that in high glucose concentrations the AP generation relies only on L-type voltage-gated Ca^2+^ channels activity [25]. Besides its contribution to the electrical activity, the Ca^2+^ influx conducted by the high voltage-gated Ca^2+^ channel (HVCC) is also crucial for vesicle fusion with the plasma membrane and insulin secretion, gene transcription and regulation, cell survival and differentiation.

## 2. Glucose-Induced Insulin Secretion (GIIS)

During a step increase in extracellular glucose concentration (usually from 3 mM to 15–20 mM glucose), insulin release follows a biphasic time course. The first phase is characterized by a fast rise and decay and relatively high amplitude. The second phase is characterized by a continuous steady release but smaller in amplitude [26,27,28]. First phase insulin release begins within 2 min after an increase of blood glucose levels, displays a distinct peak, and ends approximately 10 min later [29]. In the first phase of GIIS, the rise in intracellular Ca^2+^ concentration leads to the fusion of the pre-docked and primed insulin secretory granules from the readily releasable pool with the plasma membrane [30,31,32]. The second phase of secretion requires the recruitment, tethering, and docking of new insulin granules from a reserve pool [33] and lasts as long as glucose remains elevated with a concentration staying about two- to fivefold above the basal secretion level [34]. The insulin release lacks these two distinguishable phases and shows a progressive and monotonous increase during a ramp in extracellular glucose concentration [35]. Independent of the stimulation method, the dynamics and amplitudes of GIIS are controlled by the expression levels and biophysical properties of different HVCC α_1_ isoforms and their auxiliary subunits.

## 3. High Voltage-Gated Ca^2+^ Channel Structure

HVCCs have a hetero-multimeric composition being formed by the transmembrane α_1_, the intracellular β, extracellular α_2_δ, and transmembrane γ subunits (Figure 1A) [36,37]. In a nutshell, the α_1_ subunit forms the channel pore and confers the HVCC complex its biophysical properties and pharmacological profile, while the auxiliary subunits are important for α_1_ membrane incorporation and correct localization as well as modulation of its biophysical properties [38,39]. Additionally, at least for α_1_, α_2_δ, and β subunits, it has been shown that they also act as important scaffolds for a multitude of other regulatory and effector proteins [40]. In the next paragraphs, we will discuss the role of each HVCC subunit isoform expressed in pancreatic β-cells on electrical activity and insulin release.

## 4. α1 Subunit

The α_1_ subunit is a polypeptide comprising of four homologous repeats (I–IV) with six transmembrane segments each (S1–S6) (Figure 1B). The S1 to S4 segments of each repeat form the voltage sensing domains (VSDs) while S5–S6 loop together with S5 of all repeats come together and form the actual channel pore [36,41]. The S4 segments contain four to five positively charged amino acids evenly spaced every three amino acids that serve as voltage sensors [42].

Following the membrane depolarization, the S4 segments rotate and slide through the membrane [44,45,46], pulling on the intracellular S4–S5 linker that induces a conformational change of the S5 and S6 domains. The concerted conformational changes in all four repeats result in the opening of the channel pore [42], with each of the four VSDs having an uneven and isoform-specific contribution [23,44,46,47,48,49,50,51,52,53]. Seven genes encode for the HVCC α_1_ subunit isoforms. Based on their biophysical and pharmacological properties, the HVCC α_1_ subunits can be separated in L-type Ca^2+^ channels (Ca_V_1.1, Ca_V_1.2, Ca_V_1.3 and Ca_V_1.4), P-Q type (Ca_V_2.1), N-type (Ca_V_2.2) and R-type (Ca_V_2.3). While Ca_V_1.1 and Ca_V_1.4 seem to be exclusively and alone expressed in skeletal muscle [51] and retina, respectively [54], other excitable cells express multiple HVCC α_1_ subunit isoforms. Accordingly, mRNA profiling has demonstrated that mouse and human pancreatic β-cells express *CACNA1C* gene encoding for Ca_V_1.2-α_1C_ subunit, *CACNA1D* encoding for Ca_V_1.3-α_1D_, *CACNA1A* encoding for Ca_V_2.1-α_1A_, *CACNA1B* encoding for Ca_V_2.2-α_1B_, and *CACNA1E* encoding for Ca_V_2.3-α_1E_ subunit. Pharmacological dissection demonstrated that ~60–70% of β-cell Ca^2+^ influx is conducted by isradipine-sensitive L-type Ca_V_1 channels, ~20% by SNX-482-sensitive R-type channels, and ~20% by ω-agatoxin IVA-sensitive P/Q-type channels [55,56,57,58]. In a very limited number of studies, it has been shown that ω-conotoxin GVIA, a Ca_V_2.2 antagonist, blocks a very small percentage (~5%) of β-cell Ca^2+^ influx in both mouse and human islets [16,59]. However, since this had no effect on insulin secretion the contribution of Ca_V_2.2 N-type channels to β-cell function will not be further discussed in this review.

## 5. Ca_V_1.2 and Ca_V_1.3 L-Type Ca^2+^ Channels

L-type Ca^2+^ channels take their name from their property to conduct long-lasting and slow inactivating Ca^2+^ currents. In muscle cells, L-type Ca^2+^ channels are responsible for excitation-contraction coupling [60,61], while in most neurons, L-type calcium channels are located on the cell body and proximal dendrites and are crucial for the synaptic signal transmission and regulation of gene expression [62]. Pharmacologically L-type channels are characterized by their high sensitivity to dihydropyridine (amlodipine, felidipine, nifedipine, isradipine), phenylalkylamine (verapamil), and benzothiazepine (diltiazem) [22]. Although the binding sites for the different L-type channel blockers overlap [63,64,65], their mechanism of action is different; the dihydropyridines (DHPs) prefer and stabilize the inactivated state of the channel [66,67,68,69,70], while binding of phenylalkylamines and benzothiazepines is favored by the open channel state [71]. Consequently, the DHPs are better at blocking the channels during prolonged membrane depolarization, even close to resting membrane potential, while phenylalkylamines and benzothiazepines lead to a pronounced frequency- or use-dependent inhibition of the AP firing [71]. Consistent with this notion, early pharmacological studies in rodents have demonstrated that nifedipine has a stronger effect on β-cell electrical activity compared to verapamil and increasing the glucose concentration augments the β-cell sensitivity to verapamil but not nifedipine [25]. Despite the different activity-induced drug sensitivity, both compounds reduced the β-cell spike frequency in a dose-dependent manner, and at high concentrations (nifedipine > 10^−7^ M, verapamil > 10^−6^ M), lead to the disappearance of spikes through a decrease in amplitude [25]. Consequently, in a dose-dependent manner, verapamil reduces both phase of GIIS triggered by high glucose, and at low µM range, almost completely blocked GIIS [72] (Figure 2B,C). Importantly, 10 µM isradipine completely abolished the electrical activity and insulin secretion induced by 20 mM glucose also in human pancreatic β-cells and islets [16]. In agreement with all these in vitro pharmacology studies, overdose of L-type channel blockers in humans leads to hyperglycemia due to hypoinsulinemia [73]. However, at therapeutic doses, L-type channel blockers show no evident block of insulin release but instead help preserve β-cell mass in adults with recent-onset type1 diabetes [74]. Whether this effect is caused only by reducing the hyperglycemia-induced β-cell electrical activity and subsequently the oxidative stress or it is a direct effect of the different L-type channel isoforms inhibition remains to be shown. As previously mentioned, at mRNA level pancreatic β-cells express two L-type Ca^2+^ channels—Ca_V_1.2 and Ca_V_1.3—and they seem to have very specific roles in β-cell function and survival. Despite the fact that several transgenic mouse models for both Ca_V_1.2 [57,58,75] and Ca_V_1.3 [30,76,77] have been previously investigated, the exact role of these two isoforms on β-cell electrical activity, insulin release and β-cell mass are still controversial. Ca_V_1.2 β-cell-specific deletion reduced by ~45% Ca^2+^ influx but this has been reported to have only moderate effects on electrical activity induced by 10 mM glucose. Considering that in 10 mM glucose the β-cell membrane potential during a depolarizing plateau is ~−50 mV, the AP generation most probably relies on Na_V_ channel activity. In higher glucose concentrations (15–20 mM), when the plateau potential reaches ~−40 mV (which drives the Na_V_ channels into inactivation) the electrical activity in Ca_V_1.2^−/−^ β-cells might be initiated by another HVCC isoform. However, the P/Q-type channels activate at higher membrane potentials, making it very unlikely that they are capable of initiating the electrical activity [78,79]. While the Ca_V_2.3 R-type channels activate at lower membrane potentials, they display a very strong steady-state inactivation and low availability already at −50 mV, making them a very unlikely candidate for pace-making [80]. In contrast, the Ca_V_1.3 channels have a depolarization threshold at ~−50 mV [75,77,81,82,83], and it has been shown that they are involved in the initiation of the electrical activity in the sino-atrial node [84] and chromaffin cells of the adrenal gland medulla [85,86]. Two transgenic Ca_V_1.3 knock-out mouse models have previously been independently generated producing contradictory results regarding the involvement of Ca_V_1.3 in β-cell function. In one mouse model, Ca_V_1.3 genetic ablation did not change the β-cell Ca^2+^ current biophysical properties and did not cause a diabetes phenotype [30,77]. Additionally, L-type channel blocker isradipine and activator BayK8644 failed to alter the HVCC Ca^2+^ currents recorded in β-cells isolated from mice expressing a Ca_V_1.2 channel insensitive to dihydropyridines (Ca_V_1.2DHP^−/−^) [75]. Corroborated, these studies made a very strong case that the Ca_V_1.3 channel is not expressed in mouse β-cells. However, a second Ca_V_1.3^−/−^ mouse model did show a diabetes phenotype due to reduced postnatal β-cell generation and lower β-cell mass [76]. β-cell Ca^2+^ influx was also not reduced in amplitude, but the voltage dependence of activation was shifted towards more positive potentials by ~10 mV (Figure 2B). Given that Ca_V_1.3 L-type channels have an ~20mV more negative half maximal activation compared to Ca_V_1.2 channels, the similar whole-cell Ca^2+^ current, but a rightwards shift in the voltage dependence of activation in Ca_V_1.3^−/−^ β-cells compared to wild-type could be indicative of a compensatory upregulation of Ca_V_1.2 channels. Of note is the observation that Ca_V_1.3 deletion reduced the insulin secretion only in lower glucose concentrations, supporting the notion that Ca_V_1.3 is involved in β-cell pace-making at lower resting membrane potentials (Figure 2C) [76]. One possible explanation for the observed discrepancies between the two mouse models is that the β-cells are an inhomogeneous population with only a subset of cells expressing Ca_V_1.3 channels. In fact, a recent study has demonstrated using single channel recordings that only ~20% of β-cells express Ca_V_1.3 channels [87]. Additionally, it has been shown that Ca_V_1.3 channels have a lower dihydropyridine sensitivity compared to Ca_V_1.2 [88]; therefore, some of the pharmacology experiments might have underestimated the contribution of Ca_V_1.3 to the total β-cell Ca^2+^ influx.

Besides their contribution to electrical activity and increase in cytosolic Ca^2+^ concentration, HVCCs are also critical for vesicle fusion with the plasma membrane. At the central and peripheral nervous system synapses, Ca_V_2.1 and Ca_V_2.2 channels in particular mediate neurotransmitter vesicle release [40], whereas ribbon synapses of the auditory system and the retina utilize Ca_V_1.3 and Ca_V_1.4 L-type Ca^2+^ channels, respectively [89,90]. In neurons, exocytosis of synaptic vesicles is generally regulated by the interaction of the active zone proteins Rab3, RIM (Rab3-interacting molecules) and RBP (RIM binding proteins), and the SNARE proteins syntaxin, SNAP25 and synaptobrevin [91]. Ca^2+^ influx through the Ca_V_2 family triggers vesicle fusion with the plasma membrane through interaction with synaptotagmin and the synprint site of the II-III loop of the Ca_V_2.1 and Ca_V_2.2 channel [37,40,91]. While accumulating evidence shows that the release machinery in pancreatic β-cells has largely the same components as in the neuronal synapses, the HVCC isoform central for insulin vesicle exocytosis is Ca_V_1.2 L-type Ca^2+^ channel (Figure 2A) [30,57,92,93]. Ca_V_1.2 deletion in mice strongly reduced and delayed the first rapid phase vesicle exocytosis, indicating that Ca_V_1.2 is directly coupled to vesicle release machinery [57]. Importantly, this association is disrupted in β-cells from Type 2 Diabetes Mellitus (T2DM) patients and INS-1 cells cultured in fatty acids that mimic the diabetic state. This dissociation leads to a dramatic reduction in rapid Ca^2+^ influx-dependent exocytosis, explaining why the first phase insulin release is missing in T2DM patients [92].

Despite several decades of research, the exact molecular mechanisms for how L-type channels control β-cell function, differentiation, and survival still need further investigation. Nevertheless, it is clear that both Ca_V_1.2 and Ca_V_1.3 are intimately involved in GIIS in both mice and men. Ca_V_1.2-α_1C_ and Ca_V_1.3-α_1D_ main subunit deletion in mice led to impaired glucose tolerance and diabetes [57,76] while L-type pharmacological block showed strong effects on β-cell activity [25,72,73]. Strong evidence regarding the involvement of both L-type channel isoforms in human β-cell function comes from recent identification of genetic alteration in *CACNA1C* and *CACNA1D* genes. Ca_V_1.2 gain-of-function mutations lead to hyperinsulinism accompanied by intermittent hypoglycemia leading to death [94]. Similarly, *CACNA1D* gain-of-function mutations lead to hypoglycemic events [95,96] while genetic polymorphisms, suspected to lead to Ca_V_1.3 loss-of-function, correlate with a higher T2DM incidence [97]. However, the effect of Ca_V_1.2 and Ca_V_1.3 mutations on glucose metabolism is probably more complex and expands beyond their role in pancreatic hormone release. Both channel isoforms are expressed in many endocrine cells and are responsible for the secretion of other hormones intimately involved in glucose homeostasis such as catecholamines [98] and glucocorticoids [99,100,101].

## 6. Ca_V_2.3 R-Type Calcium Channels

Besides Ca_V_1.2 and Ca_V_1.3 L-type channels, mouse pancreatic β-cells also express Ca_V_2.3 R-type channels. Initially, Ca_V_2.3 channels have been described as having biophysical properties very similar to low voltage-activated Ca^2+^ channels (LVCC) [79,102]. Ca_V_2.3 display transient currents with fast activation and inactivation kinetics and, similar to LVCCs, show a larger Sr^+^ single-channel conductance compared to Ca^2+^ or Ba^2+^. Initially it has also been proposed that Ca_V_2.3 channel’s membrane incorporation and biophysical properties are less modulated by auxiliary β and α_2_δ HVCC subunits [79], another LVCC characteristic. However, several later studies have disproved these observations. In heterologous cells, the presence of an α_2_δ subunit does not seem to be required for Ca_V_2.3 membrane incorporation but it accelerates the activation kinetics and shifts the voltage dependence of activation towards more positive potentials [80,103,104]. By contrast, the presence of a β subunit dramatically increases Ca_V_2.3 Ca^2+^ influx by increasing the number of channels functionally incorporated in the membrane, enhancing the open probability as well as by shifting the voltage dependence of activation towards more negative potentials [80]. However, perhaps the most physiologically relevant effect of β subunits on Ca_V_2.3 function is the modulation of Ca_V_2.3 inactivation properties in a β isoform-specific manner. Depending on the co-expressed β isoform, the Ca_V_2.3 Ca^2+^ current half maximal steady-state inactivation varies from ~−50 mV to ~−80 mV [80]. What does this mean for β-cell function? At low glucose concentrations, when the membrane potential is close to −80 mV, most of the Ca_V_2.3 channels should be available to open and contribute to electrical activity and cytosolic Ca^2+^ concentration. In higher glucose concentrations (20 mM), when the resting membrane potential can reach ~−40 mV, the number of Ca_V_2.3 channels available to open will be dramatically reduced (potentially zero) depending on which β subunit they complex with. Indeed, insulin release stimulated by 20 mM glucose was completely unaffected by SNX-482 [16], a Ca_V_2.3 channel blocker [105], supporting this notion. However, SNX-482 also had minimal effects on human β-cell Ca^2+^ influx when the resting membrane potential was −70 mV [16], indicating either that Ca_V_2.3 channel expression in human β-cells is very low or that a large Ca_V_2.3 channel population is already in the inactivated state at resting membrane potential. Despite the observation that Ca_V_2.3 channels have a marginal contribution to β-cell functions in humans, several polymorphisms in *CACNA1E* gene encoding for Ca_V_2.3-α_1E_ subunit have been reported to associate with an increased incidence of T2DM [106,107]. This could be caused by effects of Ca_V_2.3 on insulin release in low glucose concentrations or functions outside the pancreas. Supporting the extra-pancreatic hypothesis regarding the role of Ca_V_2.3 channels in glucose homeostasis come the observations that mice with a global genetic deletion of Ca_V_2.3 channel [108] develop a non-insulin-dependent form of diabetes with increased body weight and fasting glucose levels, and impaired insulin sensitivity accompanied by higher basal insulin secretion [109]. A second independently generated mouse model also showed impaired glucose tolerance but, in this case, caused by reduced insulin secretion [110]. Electrophysiological analysis of pancreatic β-cells isolated from Ca_V_2.3^−/−^ mice showed an ~25% reduction in whole-cell Ca^2+^ influx [55], similar to the pharmacological effect of SNX-482 application observed in other studies (Figure 2B) [56,57]. This reduction in Ca^2+^ influx did not affect the peak islet Ca^2+^ transients induced by 15mM glucose [55]. However, after the initial peak, the Ca^2+^ transients showed a reduced amplitude and oscillation frequency in Ca_V_2.3^−/−^ mice compared to control. Similarly, the first-phase GIIS in response to 16.7 mM glucose was neither affected by Ca_V_2.3 genetic ablation nor SNX-482 application but the second phase was strongly reduced (Figure 2C). Interestingly, capacitance measurements showed that the initial Ca^2+^ influx-induced exocytosis was not altered by Ca_V_2.3 deletion, in good agreement with the observations that insulin vesicle release in mouse pancreatic β-cells is primarily controlled by Ca_V_1.2 channels [57,92]. However, the same depolarization train showed a reduced exocytosis in Ca_V_2.3^−/−^ β-cells towards the end of the train, indicating that Ca_V_2.3-mediated Ca^2+^ influx is required for sustained vesicle release. The reduced exocytosis during prolonged depolarizing trains in combination with reduced electrical activity (as indicated by reduced oscillation frequency of the islet Ca^2+^ transients) result in the diminished second phase GIIS and ultimately glucose intolerance. However, Ca_V_2.3 deletion could also alter glucose metabolism by controlling the release of the other islet hormones. Cav2.3-deficient mice showed an impaired glucose suppression of glucagon release [111]. However, this is not caused by a direct effect on α-cell glucagon release [55] but indirectly through a reduced somatostatin secretion [112,113], a hormone known to have an inhibitory paracrine effect on insulin and mostly glucagon secretion at high plasma glucose concentrations [3,4,114,115].

## 7. Ca_V_2.1 P/Q- Type Ca^2+^ Channels

Although the primary role in insulin release is taken by the L- and R-type Ca^2+^ channels, pharmacological dissection shows that approximately 20% of Ca^2+^ influx in pancreatic β-cells of both mouse [57] and human [16,116] is conducted by ω-agatoxin IVA-sensitive Ca_V_2.1, P/Q-type channels. The P/Q-type Ca^2+^ currents activate fast, with a depolarization threshold of ~−30 mV, and reach their maximum at +10 mV. Although experimental evidence is missing, the biophysical properties Ca_V_2.1 channels suggest that they do not directly account for the electrical activity but contribute to the increase in cytosolic Ca^2+^ concentration during the depolarization train. In agreement with this hypothesis, ω-agatoxin IVA application reduced insulin release stimulated by 6 mM and 20 mM glucose albeit to a lot lower extent compared to L-type channel blockers [16,116]. It has been proposed that Ca_V_2.1 channels play a much more important role in human β-cells compared to mouse. While in mouse it has been shown that Ca_V_1.2 channels are coupled to insulin vesicle exocytosis [57], in human β-cells ω-agatoxin IVA resulted in a much stronger reduction in exocytosis of immediately releasable pool of granules compared to isradipine [16]. This suggests that in human β-cells the vesicle release machinery is centered around Ca_V_2.1 channel (Figure 2A). However, a very elegant recent study contradicted these observations [92]. Using high-resolution microscopy, Gandasi and colleagues [92] showed that Ca_V_1.2 L-type Ca^2+^ channels cluster with the insulin vesicles both in the INS-1 rat insulin secreting cell line as well as in human β-cells. They showed that the interaction occurs via the Ca_V_1.2 II–III loop with the Munc13 synaptic protein and, importantly, the disruption of this interaction abolished the fast exocytosis. Interestingly, T2DM-mimicking conditions also disturb the association of Ca_V_1.2 L-type channels with the insulin vesicles underlying the blunted first phase insulin release observed in T2DM patients. Additionally, many loss- and gain-of-function mutations in the *CACNA1A* gene encoding for Ca_V_2.1-α_1A_ subunit have been identified in humans as a cause of epilepsy, episodic ataxia, and migraine [117]. However, to our knowledge, none of these mutations have been associated to hyperglycemia or hyperinsulinism, cementing the notion that Ca_V_2.1 is not critical for human β-cell function and insulin release.

## 8. β-Subunits

The HVCC complex cannot be formed by α_1_ subunits alone as proper membrane trafficking and localization depends on the auxiliary subunits, among which the intracellular β play a central role. It has been well established that the β subunit promotes the membrane surface expression of Ca_V_1 and Ca_V_2 channels. Initially, it was postulated that the β-subunits enhance channel trafficking by masking an endoplasmic reticulum retention signal at the α_1_ I–II intracellular linker [118]. Such a mechanism could not be confirmed in subsequent experiments [119,120,121]. Regardless of the mechanism, the notion that a β subunit is required for α_1_ membrane incorporation was cemented by many studies performed in heterologous expression systems, where proper Ca^2+^ current densities could not be achieved in the absence of a co-expressed β subunit [80,118,122,123]. In congruency with these observations, the genetic ablation of the β_1_ and β_2_ subunits led to a dramatically reduced HVCC current amplitude in skeletal [124,125] and cardiac [126] muscle, neurons [127], retina [128], and inner hair cells (IHC) [129]. Conversely, β subunit overexpression increases HVCC currents in native cardiac myocytes, suggesting that HVCC membrane expression is limited by β subunit abundance [130]. The dogma that the presence of a β subunit is absolutely necessary for the membrane incorporation of the HVCC complex was, however, challenged when cardiomyocyte-specific conditional deletion of the *CACNB2* gene in adult mice reduced β_2_ protein by 96% but caused only a modest 29% reduction in Ca^2+^ current, with no obvious cardiac impairment [131]. A possible explanation for this controversial result might be that the presence of other β subunit isoforms compensates for the loss of β_2_ [132]. Alternatively, owing to its high affinity binding to α_1_ subunit and their close proximity in the ER, the remnant 4% β_2_ are sufficient to inhibit the α_1_ subunit ER retention signal and release it into the cytosol, where other α_1_ interacting proteins are responsible for proper membrane incorporation. Additional to their role in promoting α_1_ membrane incorporation, the β subunits modulate in an isoform and splice variant-specific manner the channel’s voltage dependence [80,122,124,126] and time course [122,130] of activation and inactivation. The β subunits alter the whole cell HVCC Ca^2+^ current kinetics by modulating the single-channel open probability properties [133,134,135,136,137].

The β subunits are encoded by four genes (*CACNB1-CACNB4*) with broad tissue distributions [138]. While some tissues show a very restrictive expression pattern (e.g., β_1_ in skeletal and β_2_ in cardiac muscle) other cell types such as endocrine cells and neurons express at least two different β isoforms, with the β_2_ showing the widest expression pattern. Pancreatic islet cells express both β_2_ and β_3_ subunits [139]. In contradiction with the known role of β subunits in promoting α_1_ membrane incorporation, β_3_ deletion [140] in pancreatic β-cells showed no change in HVCC Ca^2+^ current properties [139]. Instead, β_3_^−/−^ mice showed a higher insulin release and glucose tolerance due to increased glucose-stimulated intracellular Ca^2+^ oscillation frequency [139]. Under normal conditions in pancreatic β-cells, β_3_ isoform negatively modulates inositol triphosphate receptor (InsP_3_R) activity and therefore its genetic deletion enhanced InsP_3_R function, resulting in increased Ca^2+^ mobilization from internal stores (Figure 2) [139]. In agreement with those earlier findings and using a second independently generated β_3_^−/−^ mouse model [141], recently, it has been shown that β_3_ deletion protects mice against high-fat-diet-induced diabetes, while β_3_ overexpression in isolated human islets impaired insulin secretion [142]. Importantly, the InsP_3_R-β_3_ interaction is not restricted to pancreatic β-cells and it works in a similar manner in controlling intracellular Ca^2+^ signals in fibroblasts [143]. β_3_ deletion enhanced fibroblast InsP_3_R store Ca^2+^ release and consequently enhanced fibroblast migration in vitro and wound healing in vivo [143]. The observation that in pancreatic β-cells β_3_ isoform has HVCC independent functions suggests that β_2_ isoform must be responsible for promoting HVCC membrane incorporation and biophysical properties. Knock-out of β_2_ HVCC isoform is embryonically lethal [126]. However, adult mice harboring only an extracardiac tissue deletion of β_2_ show altered retina development [128] and impaired hearing due to impaired inner hair cell development and reduced Ca^2+^ influx [129]. However, β_2_ deletion did not lead to any metabolic dysfunctions. This effect could be explained by a functional compensation by the existing β_3_ subunits. While in the presence of the β_2_ subunit, β_3_ takes HVCC-independent functions, it is possible that, upon β_2_ genetic deletion, β_3_ assumes its role as a HVCC subunit, as it does in other cell types [138,141,144].

As members of the membrane-associated guanylate kinase (MAGUK) protein family [138], β subunits are comprised of two highly conserved central domains—GK (guanylate kinase) and SH3 (Src-homology 3)—separated by a less conserved HOOK domain and flanked and interspersed by more variable N- and C-termini (Figure 1D). The β_2_ subunit is subject to extensive alternative splicing of the N-terminus and within the central HOOK domain [138]. To date, there are nine known β_2_ splice variants that confer the α_1_ subunits’ very distinct biophysical properties [138,145]. Functionally, perhaps the most interesting ones are the β_2a_ and β_2e_ variants that, besides binding to the α_1_ subunits, also interact with the plasma membrane through palmitoyl groups or electrostatic and hydrophobic interactions, respectively [138,146]. RNAseq data showed that human pancreatic islets highly express β_2a_ splice variant. β_2a_ overexpression in INS-1 rat insulinoma cells or higher palmitoylation results in an increased amount of HVCC in the membrane, leading to higher intracellular Ca^2+^ concentration that promotes apoptotic cell death [147].

The β subunits do not control only the HVCC density at the plasma membrane but also the subcellular localization. In central and sensory neurons, the neurotransmitter release is restricted to a specialized presynaptic compartment called the active zone (AZ) [91]. The AZ proteins play a critical organizational role in defining the presynaptic sites of synaptic vesicle docking and fusion. Several proteins important for the AZ organization have been characterized, including ELKS (also known as ERC1) and CAST (ERC2) [148], and both have been shown to directly interact with other AZ proteins, including RIM1, Piccolo and Bassoon, and indirectly with Munc13-1. This leads to the formation of a large molecular complex that regulates the architecture of the presynaptic compartment. Importantly, it has been shown that both CAST and ELKS physically and functionally interact with β_4_ subunit in neurons [149,150]. Although pancreatic β-cells lack defined active zones, it was shown that the insulin vesicles are polarized towards the venous vasculature [151]. ELKS binding to β_2_ subunit in pancreatic β-cells enhanced L-type Ca^2+^ current at the vascular side of the β-cell plasma membrane and regulated the initial polarized Ca^2+^ influx and rapid insulin vesicle exocytosis [152]. Another important mechanism that dynamically modulates the HVCC membrane incorporation and function is the interaction of the β and α_1_ subunits with the small GTPases RGK protein family members (Rem, Rem2, Rad, Gem/Kir) [153]. Initially it has been shown that co-expression of Gem with Ca_V_1.2 or Ca_V_1.3 channels resulted in Ca^2+^ influx inhibition, and these findings were later extended to the other RGK family members (Rad [154], Rem and Rem2 [155,156], Gem [157,158]) and HVCC isoforms [156,159]. RGK proteins have been reported to inhibit HVCC activity by distinct mechanisms, including a reduction of the number of channels in the membrane and lowering the single channel open probability [156,157]. Functionally, the different RGK family members exert their effects in a β-binding-dependent and -independent manner. Rem and Rad use both β-binding-dependent and independent mechanisms, whereas Gem and Rem2 solely utilize a β-binding-dependent mode of inhibition [160]. Importantly, the RGKs expression is activity-dependent as it has been shown that high electrical activity in response to elevated glucose levels strongly induced Rem2 expression in the mouse insulinoma MIN6 cell line [155], while in neurons, KCl treatment upregulates Rem2 and Gem mRNA levels [161]. Similarly, mice deficient for Gem GTPase show abnormal glucose tolerance and reduced intracellular Ca^2+^ transients [158]. This evidence demonstrates that the interaction between RGKs and β_2_ subunit in pancreatic β-cells and RGK-dependent modulation of HVCC function is controlled by the glucose levels.

## 9. α2δ Subunits

A total of four genes (*CACNA2D1-4*) encode for α_2_δ subunits (α_2_δ-1 to α_2_δ-4), with distinct tissue distribution [162]. Each α_2_δ subunit is a product of a single gene that is post-translationally cleaved into α_2_ and δ peptides, which remain associated via disulfide bonds [163]. Until recently, it had been considered that the δ subunit constitutes a single-pass membrane, and the α_2_ subunit a glycosylated extracellular protein. However, this classical view has been challenged by the observation that α_2_δ subunits can be GPI-anchored to the membrane and that this posttranslational modification may be crucial for α_2_δ function [164]. Few attempts have previously been made to identify the interaction site between α_2_δ and α_1_ subunit. Using coimmunoprecipitation assays of α_2_δ-1 together with trypsinized skeletal muscle α_1S_ subunit, it has been shown that the first extracellular loop of the α_1S_ IIIS5–IIIS6 linker bind to α_2_δ-1 subunit [165]. It has also been proposed that the putative transmembrane segment of the δ subunit interacts with α_1_ [166]; however, if α_2_δ subunits are GPI-anchored proteins [164], this notion needs to be reconsidered. Another interaction between α_2_δ and α_1_ subunit occurs via the common coordination of an ion in the metal ion adhesion site (MIDAS) (Figure 1C). While all α_2_δ subunits contain a putative MIDAS, only α_2_δ-1 and α_2_δ-2 incorporate the “perfect” motif, where five amino acids coordinate one metal ion [167]. The 3.6 Å resolution cryo-EM structure of the Ca_V_1.1 channel complex [36] allowed for the first time to visualize, predict, and validate the amino acids responsible for the interaction between α_1_ and α_2_δ subunits. The structure shows that α_1S_ has four possible interaction sites with α_2_δ-1 subunit located in the first three repeats confirming the previously published studies [165,166,167,168]. Functional mutagenesis confirmed [169,170] that many of these potential interactions are important for conferring the two-fold role of α_2_δ in the HVCC complex: (a) to increase the number of channels functionally incorporated in the membrane and (b) to modify the Ca^2+^ current biophysical properties. However, the effects of α_2_δ are α_1_ isoform- and tissue-specific. Due to the large number of interaction sites between the two proteins, α_1_ and α_2_δ isoform-specific sequence variations could be responsible for preferential interaction between certain isoforms as well as differential modulation. It has been shown that the α_2_δ subunit is required for membrane incorporation of Ca_V_1.2-α_1C_, Ca_V_1.3-α_1D_, Ca_V_2.3-α_1F_, Ca_V_2.1-α_1A_, and Ca_V_2.2-α_1B_ subunits expressed in human embryonic kidney (HEK) cells, *Xenopus* oocytes, or neurons [83,171,172]. However, we have shown that when expressed in muscle cells, the Ca_V_1.1-α_1S_ and Ca_V_1.2-α_1C_ channels do not require an α_2_δ for membrane incorporation [38,39,50,173,174]. This effect is most probably caused by supplementary membrane targeting interactions present in muscle cells such as the newly identified Stac3 adaptor proteins [175,176,177,178,179,180,181,182,183]. Functional evidence suggests that membrane expression of Ca_V_2.3- α_1E_ channels is also less dependent on the presence of a α_2_δ subunit. Expression of Ca_V_2.3-α_1E_ alone in *Xenopus* oocytes produced large currents [102,104] and co-expression of α_2_δ-1 failed to further increase Ca_V_2.3 currents [104]. When expressed in HEK-293 cells, α_2_δ-1 produced a twofold increase in the Ca_V_2.3-α_1E_ current density if β subunit was not present [80]. Nevertheless, if a β subunit was expressed, α_2_δ-1 failed to further increase the Ca_V_2.3 currents. This evidence strongly suggests that the Ca_V_2.3-α_1E_ subunit has weaker interactions with α_2_δ compared to the other HVCCs and might be less if at all modulated in native cells. Regarding the role of α_2_δs in modulating the Ca^2+^ current biophysical properties, we and others have shown that the Ca_V_1.2-α_1C_ channel is the only HVCC where α_2_δ induces a shift in the voltage dependence of activation towards more hyperpolarizing potentials [168,174,184,185]. We have also shown that the lack of α_2_δ-1 slows down the activation and inactivation kinetics of α_1C_ [174], and this result has been confirmed in the α_2_δ-1^-/-^ mouse model [186]. Conversely, we demonstrated that the shRNA knock-down of α_2_δ-1 in skeletal muscle cells has the opposite effects on α_1S_ current kinetics, increasing the speed of current activation and inactivation [173]. While α_2_δs do not seem to alter the voltage dependence of α_1A_ [187], α_1B_ [188], and α_1E_ [80], there is evidence that different α_2_δ isoforms alter the inactivation kinetics of α_1B_ channels in an α_2_δ isoform-specific manner [189] and α_2_δ-1 alters the activation kinetics of Ca_V_2.3 channels [104]. Therefore, genetic alterations of α_2_δ subunit isoforms will lead to complex changes of Ca^2+^ influx biophysical properties and subsequently a complex disease phenotype. However, the initial characterization showed that global deletion of α_2_δ-1 (the most widely expressed α_2_δ isoform [190]) does not lead to an obvious phenotype [186]. Mice looked normal, were viable, and without gross anatomical alterations. However, functionally, they had diminished cardiac contractility due to reduced Ca_V_1.2 Ca^2+^ influx in cardiac myocytes. In agreement with previous studies performed in heterologous expression systems [168,184,185] or muscle cells [39,174], deletion of α_2_δ-1 reduced the Ca^2+^ influx by >40%, slowed down the activation and inactivation kinetics and led to a positive shift in the voltage dependence of activation and inactivation [186]. Additionally, male α_2_δ-1^−/−^ mice displayed a tendency for bladder dilation [186] and enlarged kidneys [56]. A closer analysis indicated that α_2_δ-1 deletion in mice led to polyuria, polydipsia, and reduced survival in a sex-specific manner. Male mice displayed threefold higher basal non-fasting blood glucose levels at one month and approximately ninefold higher at five months of age. Female mice showed minimal alteration of their basal glucose levels. An intraperitoneal glucose tolerance test also showed that male mice are more affected. Two hours after the glucose challenge, male α_2_δ-1^−/−^ mice failed to reduce the plasma glucose concentration back to base levels, whereas α_2_δ-1^−/−^ females, albeit with a slower kinetic compared to wild types, did so. At the single-cell level, α_2_δ-1 deletion (the main α_2_δ isoform expressed in mouse pancreatic β-cells) equally reduced the β-cell Ca^2+^ influx by ~70% and slowed down the activation and inactivation kinetics in both males and females. Pharmacological dissection showed that all HVCC isoforms expressed in pancreatic β-cells are equally reduced, confirming the role of α_2_δ in membrane targeting of all HVCCs in a native cell system. However, despite an equal effect of α_2_δ-1 deletion on β-cell Ca^2+^ currents in both sexes, the islet Ca^2+^ transients, insulin release and β-cell mass were more reduced in males compared to females demonstrating the need for a better understanding of the role of sex-dimorphism in β-cell function and diabetes etiology [191].

Additional evidence regarding the critical role of α_2_δ-1 in glucose homeostasis comes from recent genome-wide association studies where it has been shown that variants in TCF7L2 transcription factor confer the strongest risk of T2D among common genetic variants [192]. Indirectly via ISL1, TCF7L2 regulates proinsulin production and processing via MAFA, PDX1, NKX6.1, PCSK1, PCSK2, and SLC30A8 [192]. However, TCF7L2 is also a critical regulator of α_2_δ-1 expression in pancreatic β-cells [193]. Silencing the TCF7L2 activity results in reduced α_2_δ-1 protein levels causing reduced Ca_V_1.2 membrane expression due to retention of the channel in recycling endosomes. The altered Ca^2+^ influx reduces the cytosolic Ca^2+^ transients as well as insulin release [193], as seen in the murine α_2_δ-1^−/−^ β-cells [56]. Overexpression of α_2_δ-1 in TCF7L2-silenced cells rescued the TCF7L2-dependent impairment of Ca^2+^ signaling, but not the reduced insulin secretion. This is understandable as TCF7L2 transcription factor activity is responsible also for the expression of many proteins involved in insulin exocytosis, such as Synaptotagmin-14, syntaxin binding protein 1 (Munc 18-1), and Vamp2 [193]. Therefore, TCF7L2 activity controls both insulin synthesis and release.

Although several patients carrying loss-of-function mutations in *CACNA2D1* gene encoding for α_2_δ-1 have been identified with epilepsy and intellectual disability [194], short QT syndromes [195,196] and cardiac arrhythmias [197], no metabolic alterations have been reported for these patients. Metabolic disorders and increase in body weight have been reported in patients prescribed gabapentinoids drugs (inhibiting α_2_δ-1 and α_2_δ-2 function [198]) against neuropathic pain and seizure disorders [199,200]; however, it is difficult to speculate on the mechanism of action. In vivo, gabapentin (GBP) and pregabalin (PG) application has been shown to have hardly any acute analgesic effects but to be efficient in releasing neuropathic pain upon chronic application (reviewed in [201,202]). One likely explanation for their mode of action is that chronic treatment lowers the amount of α_2_δ-1 and α_2_δ-2 on the cell surface by inhibiting rab11-dependent recycling [203], thereby decreasing the trafficking of the channel complex to the membrane and reducing Ca^2+^ influx [204,205]. However, in vitro, gabapentinoid drug application resulted in rather inconsistent effects depending on the cell system studied. Long-term incubation caused reduced Ca^2+^ currents in dorsal root ganglion neurons [205] but resulted in decreased catecholamine secretion from adrenal chromaffin cells without affecting the Ca^2+^ influx [206]. Studies using acute gabapentinoids application either report no impact at all on HVCC Ca^2+^ current density [205] or show inhibitory effects on transmitter release and/or Ca^2+^ currents [207,208]. To date, no information regarding the effect of gabapentinoid drugs application on pancreatic β-cell function has been published. Due to its very broad tissue expression pattern, alteration in α_2_δ-1 expression level and function (due to loss- and gain-of-function mutations or gabapentinoid drugs treatment) will also affect glucose metabolism independent of β-cell insulin release. It has been shown that reduced surface expression of α_2_δ-1 in the ventromedial hypothalamus (VMH) neurons leads to hyperphagia, obesity, and metabolic disturbances [209]. Interestingly, α_2_δ-1 restricted deletion in SF1^+^ (steroidogenic factor-1) VMH neurons showed a sex-specific incidence of the metabolic alterations [210]. While male mice exhibit glucose intolerance, altered lipolysis, and decreased cholesterol content in adipose tissue [211], α_2_δ-1^−/−^ females showed only a modest increase in body weight and enhanced glycemic control when fed a chow diet [210,211]. However, the glucose tolerance was altered to a lot smaller extent compared to global α_2_δ-1 deletion [56]. Mechanistically, α_2_δ-1 seems to control the activity of SF1^+^ neurons independent of its role as an HVCC auxiliary subunit. α_2_δ-1 deletion did not reduce the whole-cell Ca^2+^ influx in SF1^+^ neurons but led to a membrane hyperpolarization and dramatically reduced electrical activity and synaptic transmission [211]. In fact, several recent studies demonstrate the role of α_2_δ proteins in synapse formation and function or neuronal excitability independent of its role as an HVCC subunit [202,212,213] but due to its interactions with NMDA [214] and GABAA receptors [215], thrombospondins [216] or BK potassium channels [217]. In pancreatic β-cells, it has been shown that the interaction of α_2_δ-1 with RalA GTPase is important for binding α_2_δ-1 on insulin granules and tethering these granules to the HVCCs in the plasma membrane [218]. Therefore, α_2_δ-1 shines out as a very important modulator of glucose metabolism though its pancreatic and extra-pancreatic roles acting as an HVCC subunit or through HVCC-independent interactions.

## 10. γ Subunit

The initial purification of the skeletal muscle HVCC complex showed that besides the intracellular β_1_ and extracellular α_2_δ-1, the HVCC complex also contains the γ_1_ subunit [219,220]. Recent Cryo-EM data confirmed that γ_1_ subunit is a transmembrane protein consisting of four helices and interacts with the fourth VSD of the α_1_ subunit (Figure 1B) [36,41]. In skeletal muscle, the γ_1_ subunit has a dual role: to limit the Ca^2+^ influx by stabilizing the inactivated state of Ca_V_1.1 channel [221,222] and to reduce the depolarization-induced sarcoplasmic Ca^2+^ release [223]. γ_1_ belongs to a family of eight genes (γ_1_ to γ_8_) predominantly expressed in the brain. They are known as transmembrane AMPA receptor regulatory proteins (TARPs) [37] as they play an important role in targeting and anchoring of these ionotropic glutamate receptors in the postsynaptic membrane of neurons and glia cells [220]. However, heterologous expression systems have demonstrated that several γ isoforms can modulate HVCC function [224,225,226,227,228]. Coexpression of the γ_1_ subunit together with Ca_V_1.2 L-type Ca^2+^ channels in *Xenopus laevis* oocytes [225] or tsA201 cells [227] increased the current density and led a shift of the voltage dependence of activation towards more negative potentials. The expression of Ca_V_1.2 complex (α_1C_ + β_1_ or α_1C_ + β_2_ + α_2_δ − 1) in HEK293 cells showed that γ_4_, γ_6_, γ_7_ and γ_8_ subunits interact with the channel and modulate Ca^2+^ current properties in a γ isoform-specific manner. In the presence of β_1_ and α_2_δ-1 subunits γ_4_, γ_7_ and γ_8_ shift the voltage dependence of inactivation towards more positive potentials without affecting the voltage dependence of activation [229]. Therefore, contradictory to the effect of γ_1_ on Ca_V_1.1 L-type skeletal muscle channel, γ_1_ [225], γ_4_, γ_7_ and γ_8_ [229] seem to have a gain-of-function effect on Ca_V_1.2 L-type Ca^2+^ currents. However, all these effects were gone in the presence of the β_2_ subunit [229]. Furthermore, the results were obtained in heterologous expression systems, thereby omitting other potentially critical modulatory factors. Recently, it has been shown that γ4 subunits modulate HVCC currents in pancreatic β-cells [230]. This is the first evidence that a γ subunit can modulate an HVCC function in a native cell system besides γ_1_ and Ca_V_1.1 in skeletal muscle cells. In good agreement with the heterologous expression systems [225,227,229], reduced γ_4_ expression in pancreatic β-cells leads to decreased L-type Ca^2+^ currents and therefore glucose-stimulated insulin secretion. Interestingly, γ_4_ expression is reduced in islets from human donors with diabetes, hyperglycemic and T2D animal models Goto-Kakizaki rats, and *db*/*db* mice, as well as in islets cultured in conditions simulating diabetes such as high glucose and palmitate [230].

## 11. Conclusions

The crucial role of HVCC Ca^2+^ influx on excitability, vesicle exocytosis, and gene transcription and regulation has been extensively demonstrated in many excitable cells such as neurons, cardiac myocytes, skeletal muscle, neuro endocrine cells, etc. Critical information regarding the biophysical and pharmacological properties of all HVCC subunit isoforms has been obtained in simplified heterologous co-expression systems such as *Xenopus laevis* oocytes or HEK293 cells. In this review article, we summarized the knowledge regarding the structure and function of the HVCCs, how this impacts on pancreatic β-cell insulin release and glucose metabolism and, whenever we saw fit, compared these findings with data obtained from research performed in other cell types. The physiological role of Ca^2+^ channels subunits in pancreatic hormone release and glucose metabolism has been obtained mostly from research performed on transgenic mouse models. Critical experiments have also been performed in isolated human pancreatic islets and cells from healthy or diabetes donors. This provides a very important translational aspect to the research performed in animal models.

Genetic ablation of most HVCC subunits in pancreatic β-cells led to reduced β-cell Ca^2+^ influx and therefore altered excitability [56,57], reduced intracellular Ca^2+^ concentration and oscillations [55,56], reduced insulin secretion [55,56,57,230], and also altered β-cell differentiation and survival [55,56,76]. However, the severity of the effects induced by HVCC subunit deletion varied greatly. Ca_V_1.2 genetic deletion or pharmacological block reduced the β-cell Ca^2+^ influx by ~60% (Figure 2C, dark blue) compared to controls (grey) and, because Ca_V_1.2-α_1C_ subunit is coupled to vesicle release (Figure 2A) it almost completely abolished the first phase insulin release and strongly reduced second-phase release (Figure 2C, dark blue). Ca_V_1.3 deletion did not affect the β-cell peak Ca^2+^ influx but resulted in a shift of the voltage dependence of activation towards more positive potentials. Ca_V_1.3 deletion reduced insulin release at lower glucose concentration and, due to smaller β-cell mass, also reduced the total insulin release, thereby leading to glucose intolerance in mice. Ca_V_2.1 pharmacological block resulted in an ~20% reduction in β-cell Ca^2+^ currents. Although its role on GIIS has been demonstrated only in static experiments, based on its potential contribution to the peak of APs, it is expected that Ca_V_2.1 loss-of-function would slightly reduce both phases of GIIS. If in human β-cells Ca_V_2.1 is coupled to secretory machinery, then its genetic deletion or pharmacological block will reduce both phases of GIIS to a similar extent as Ca_V_1.2 does in mouse β-cells. Similar to Ca_V_2.1, Ca_V_2.3 channels also contribute with ~20% to total β-cell Ca^2+^ influx. Ca_V_2.3 deletion or SNX-482 application did not alter the first phase but strongly reduced second-phase GIIS due to its proposed role in promoting vesicle trafficking to the release sites (Figure 2, cyan). α_2_δ-1, the main α_2_δ isoforms expressed in pancreatic β-cells, forms a complex with all HVCC isoforms. Consequently, its genetic deletion resulted in ~70% decreased β-cell Ca^2+^ influx and a strong reduction of both phases GIIS (Figure 2, red). γ_4_ subunit seems to promote only L-type channel membrane incorporation; therefore, its reduced expression is expected to have a similar effect on GIIS as L-type channel block (Figure 2, yellow). An exception from the rule applies. In pancreatic β-cell, the β_3_ subunit has HVCC-independent functions and its genetic deletion increases IP_3_R Ca^2+^ release from intracellular stores, thereby most probably increasing insulin secretion only in the second phase since the first phase is limited by the size of the readily releasable pool of vesicles (Figure 2, green).

During the preparation of Figure 2, we intended to incorporate the information regarding the role of all HVCC subunits on glucose-induced electrical activity and intracellular Ca^2+^ transients. However, such a figure would be mostly speculative since the experiments have either never been performed or the data have been obtained under very different experimental settings. We are confident that these gaps in our knowledge will soon be filled and complemented by additional RNAseq and proteomics data. This will shed light on the known and hitherto unknown roles of the HVCC subunits in pancreatic hormone release and glucose homeostasis in health and diabetes.

## Figures and Tables

**Figure 1 cells-10-02004-f001:**
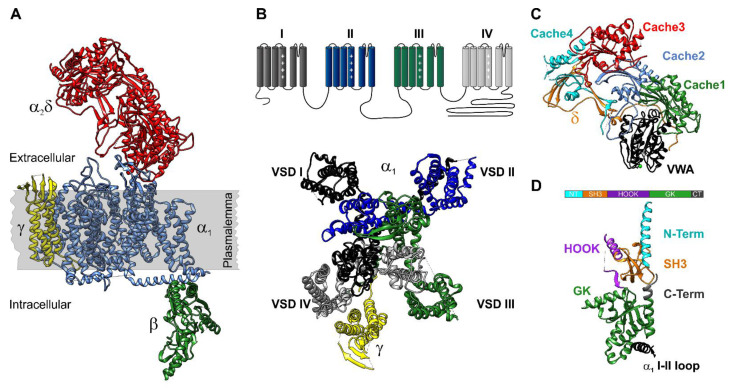
Structure of the HVCC complex. (**A**). The HVCC complex consists of the transmembrane, pore-forming α_1_ and auxiliary intracellular β, extracellular α_2_δ and transmembrane γ subunits (PDB access code 5GJV [36]). (**B**). The α_1_ subunit is formed by four homologous repeats with six transmembrane domains each. The S1 to S4 segments of each repeat form the voltage sensing domain (VSD). S4 carrying four to five positively charged amino acids lysine or arginine serve as actual voltage sensors. The S5 and S6 segments of each repeat form the channel pore. The transmembrane γ subunit interacts with the fourth VSD. (**C**). The extracellular α_2_δ subunit consists of four Cache and one von Willebrand factor A (VWA) domain. The VWA domain contains the MIDAS motif (metal ion adhesion site) that coordinates a metal ion (Ca^2+^) together with residues in the extracellular linker between S1 and S2 segments of the first VSD. (**D**). Topology of the β_2a_ subunit in complex with the Ca_V_1.2-AID (α interaction domain) located in the intracellular loop between repeats I and II (PDB access code 5V2P [43]). The β subunits are organized in 5 regions: N-terminus, the SH3 domain, HOOK region, and the GK domain.

**Figure 2 cells-10-02004-f002:**
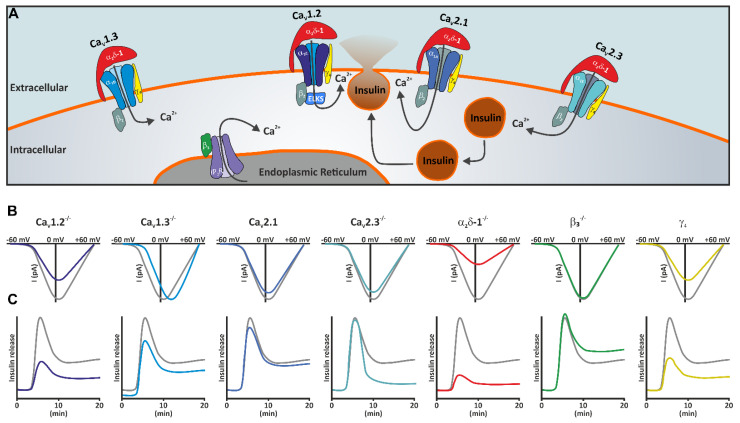
Role of the HVCC subunits on pancreatic β-cell insulin release. (**A**). Localization of the HVCC isoforms in pancreatic β-cells with respect to insulin vesicles. At least in mouse, Ca_V_1.2 channels are coupled to vesicle release, while evidence suggests that in human β-cells, Ca_V_2.1 takes the central role. The localization of Ca_V_1.2 channels at the release sites is controlled by the interaction between β_2_ subunit and ELKS. Ca_V_2.3 channels are necessary for sustained insulin release, while the role of Ca_V_1.3 is very controversial. β3 isoform does not come in complex with any HVCC but interacts with the IP_3_R. (**B**) Effect of genetic deletion or pharmacological block of each HVCC subunit isoform on total β-cell Ca^2+^ current. Ca_V_1.2 contribute with ~45% of the whole-cell Ca^2+^ influx while Ca_V_2.1 and Ca_V_2.3 with ~20% each. Ca_V_1.3 deletion only shifts the voltage dependence of activation. α_2_δ-1 deletion reduces all HVCC Ca^2+^ currents by ~70% while γ_4_ altered expression reduces only L-type Ca^2+^ currents (~70%). (**C**) Role of the HVCC subunits on GIIS. Ca_V_1.2 deletion has the strongest effect as it is coupled to insulin vesicle release. Ca_V_1.3 deletion reduces insulin secretion at lower glucose concentration while lack of Ca_V_2.3 Ca^2+^ currents reduces second-phase GIIS. α_2_δ-1 deletion strongly reduces GIIS, while β_3_ deletion increases insulin release. γ_4_ reduced expression is expected to have similar effects as Ca_V_1.2 deletion.

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
