# Peer review of "Role of High Voltage-Gated Ca2+ Channel Subunits in Pancreatic β-Cell Insulin Release. From Structure to Function"

_cells, 2021, doi:10.3390/cells10082004_

Round 1

Reviewer 1 Report

Tuluc et al. present a cutting-edge, well-balanced and clear overview of the role of Ca channels and their protein subunits in insulin release.  The authors consider the current literature including their own ground-breaking results.

I have only minor questions, the answers to which may be of interest to a wider readership, which would be desirable for this review article:

1) The authors discuss the sensitivity of voltage-gated calcium channels to calcium channel blockers in beta-cells. In this context, are there data on the sensitivity of Cav1.2 and Cav1.3 to verapamil? Is there an equal sensitivity to verapamil or is the sensitivity different between the two channels?

2) It would also be interesting to briefly elaborate to what extent therapeutic doses of widely used calcium channel blockers such as verapamil (indicated for supraventricular tachycardia) and amlodipine (hypertension) increase patients' susceptibility to diabetes?

3) Is it known whether pregabalin and gabapentin, which bind to α2δ1 and α2δ2, thereby inhibit Ca currents under experimental conditions?

4) According to Mastrolia et al. (ref. 55), loss of α2δ1 increases susceptibility to diabetes in mice. Are there any data available on whether pregabalin and gabapentin administered to patients with neuropathic pain or seizures, increase susceptibility to diabetes in these patients in addition to increasing body weight?

5) It was originally postulated that the β-subunits enhance calcium channel trafficking by masking an endoplasmic reticulum retention signal at the α1 I-II linker (Bichet D, et al. Neuron. 2000;25:177-190). Such a mechanism could not be confirmed in subsequent experiments (Cornet et al. (2002; Eur J Neurosci 16:883-95; Altier C, et al. Nat Neurosci. 2011;14:173-180; Fang K, Colecraft HM. J Physiol (Lond). 2011;589:4437-4455). Could be added by the authors.

6) For the "quiescent, activated and inactivated conformational states", the authors could in addition refer to R.W. Tsien (Annu Rev Physiol 1983, 45:341-358), who was one of the first to describe this concept for voltage-gated calcium channels in 1983. There was also very early evidence that dihydropyridines bind to and stabilise the inactivated state, e.g. by Lee and Tsien (1983; Nature 302:790-794) and Hess, Lansman, Tsien (Nature .1984311:538-544); could be added.

7) The authors mention that two Cav1.3 KO mouse strains (ref. 73 and ref. 74), produced independently, gave conflicting results regarding the involvement of Cav1.3 in β-cell function. The Cavβ3 KO mouse strain described in Ref. 134 was generated by Namkung et al. (PNAS (1998) 95:12010-12015). This mouse strain is not identical to the Cavβ3 KO strain produced independently by Murakami et al. (2002) JBC 277:40342-40351) and described in refs 133 and 135. In contrast to the phenotypes of the two Cav1.3 KO lines, the two independently produced Cavβ3 KO strains share very similar β-cell phenotypes.

8) In the 10th paragraph, the authors refer to gamma 4 and the interesting article by Luan et al. (2019, Commun Biol): are the authors aware of further data from other research groups showing that gamma4 is indeed a Ca channel subunit and not a transmembrane AMPAR-regulating protein (TARP), or is gamma 4 both?

Reviewer 2 Report

This review summarizes the knowledge regarding the structure and function of the high voltage gated Ca2+ channels (HVCCs) and their role in pancreatic beta cell insulin release and glucose metabolism. The review looks good but there are some points that should be addressed as follows:

  • The authors should add a sperate section about the “role of high voltage gated Ca2+ channels (HVCCs) in diabetes”.
  • It is better to add a table to summarize different subtypes of HVCCs and their expressions and functions in islet (alpha, beta, delta) cells.
